# Brain microdialysis to assess trace elements dynamics in traumatic brain injury: An exploratory study

Adriano Bernini[1], Sébastien Lenglet[2], Mette M. Berger [ID][3], Samia Abed-Maillard[4], Roy Thomas Daniel[5], Mahmoud Messerer[5], Mauro Oddo[6], Jean-Daniel Chiche[3], Marc Augsburger[2], Nawfel Ben-Hamouda [ID][3]*

1 Department of Clinical Neurosciences and NeuroDigital@NeuroTech, Lausanne University Hospital (CHUV), Lausanne, Switzerland, 2 Unit of Forensic Toxicology and Chemistry, University Center of Legal Medicine, Lausanne-Geneva, Geneva University Hospital and University of Geneva, Geneva, Switzerland, 3 Department of Adult Intensive Care Medicine, Lausanne University Hospital (CHUV) and University of Lausanne, Lausanne, Switzerland, 4 Clinical Research Center, Lausanne University Hospital (CHUV), Lausanne, Switzerland, 5 Department of Clinical Neuroscience, Service of Neurosurgery, Lausanne University Hospital (CHUV) and University of Lausanne, Lausanne, Switzerland, 6 Directorate for Innovation and Clinical Research, Lausanne University Hospital (CHUV), and University of Lausanne, Lausanne, Switzerland

* nawfel.ben-hamouda@chuv.ch

## Abstract

### Background

Trace elements (TEs) status alterations in the brain have been linked to neurodegenerative diseases. However, data on TEs in living humans and in the post-traumatic conditions are scarce. Some TEs (copper – Cu, selenium – Se, zinc – Zn) are involved in essential antioxidant defence. This study aims to measure the evolution of TEs concentrations in the brain and serum of severe traumatic brain injury (TBI) patients over time.

### Methods

Twenty adult patients with severe TBI were monitored using cerebral microdialysis (CMD) and blood sampling within three days of intensive care unit admission. TEs levels were measured using inductively coupled plasma system coupled to mass spectrometry.

### Results

TEs concentrations of chromium – Cr, Cu, cobalt – Co, manganese – Mn, molybdenum – Mo, Se, and Zn were quantified in brain interstitial fluid and serum. While serum and CMD levels did not differ significantly for Co, Mo and Mn, and modest differences was observed for Cr and Zn, significant differences were observed for Cu

**Data availability statement:** All relevant data are within the paper and its Supporting Information files.

**Funding:** The author(s) received no specific funding for this work.

**Competing interests:** The authors have declared that no competing interests exist.

and Se with higher serum levels (8–10-fold higher) compared to CMD. No correlation was found between serum and brain TEs levels, except for Mo.

## Conclusion

This study provides novel TEs concentration data in living TBI patients, the largest differences between brain and serum being observed for Cu and Se, serving as a basis for further research on TEs dynamics in acute brain injury.

## Introduction

Trace elements (TEs) status in the brain were already shown back in 1975 [1]. Zinc (Zn) is the most abundant TE in the central nervous system, and its status alterations combined with copper (Cu) have been associated with poor brain development and neurological recovery in children, and adults in different neurological diseases in relation to their antioxidant functions [2,3]. Indeed, some studies have investigated the role of TEs in brain function and metabolism [4] with the aim to demonstrate a correlation between brain TEs concentrations and the onset of neurodegenerative diseases or dementia [5–7]. Moreover, experimental studies showed critical roles of TEs in neuroprotection, in the context of traumatic brain injury (TBI). In this context, TEs act on oxidative stress, inflammation, apoptosis, and mitochondrial function (Table 1). Although analytical techniques have improved over time, all human samples previously studied were taken from post-mortem brains [5]. To date, data on physiological or pathological concentrations of TEs in living humans are lacking, and this is particularly true after acute brain injury (ABI).

Cerebral microdialysis (CMD) enables direct cerebral metabolic monitoring in the brain-injured patient in the intensive care unit (ICU), particularly in the context of ABI such as TBI [8]. In practice, it enables direct regional measurements of the brain interstitial tissue concentrations of main cerebral energy metabolites, including glucose, lactate, glutamate, and glycerol [8].

Excitotoxicity occurring in response to brain insults is highly involved in the outcome of patient with TBI. Post trauma mechanisms generate oxidative stress, an imbalance between oxidant and antioxidant agents that can result in neural dysfunction and neuronal death [9]. Among TEs, selenium (Se) has been shown to reduce oxidative stress via the antioxidant enzyme glutathione peroxidase activity and thus to have a neuroprotective effect [10,11]. Studies in rodents confirmed it by showing an improved neurological outcome after its administration [11,12].

In clinical studies, benefits of early Se supplementation were reported in a large study including 307 patients with severe TBI, with a reduction of the risk of unfavourable functional outcomes [13]. Zn, also involved in antioxidant defense, appears to have a neuroprotective proprieties in the aftermath of the trauma (at one month) [2,14,15]. A randomized trial including 68 head-injured patients tested Zn administration and showed significantly higher motor response scores in the Zn-supplemented group on days 15 and 21 than in the control group [16]. Iron (Fe), Cu, and Zn, have

**Table 1. Main neuroprotective pathways of trace elements in brain injury.**

| Trace element | Role | Pathways | References |
|---|---|---|---|
| **Copper (Cu)** | Antioxidant defense | Cofactor for Cu/Zn-superoxide dismutase (SOD). | [17–20] |
| | Vascular recovery | Stimulation of factors involved in vessel formation and maturation, such as VEGF. | |
| **Magnesium (Mg)** | Reduces excitotoxicity | Blocking NMDA channels and voltage-gated calcium channels. | [21,22] |
| | Increasing cerebral blood flow | Relaxation of vascular smooth muscle. | |
| | Antiapoptotic role | Upregulation of the tumor-suppressor gene TP53. | |
| | Anti-edema Effects | Downregulate aquaporin-4 channels. | |
| **Selenium (Se)** | Reduction of oxidative stress in the brain, neutralization of lipid peroxidation and antiapoptotic effect | Glutathione peroxidase (GPx) enzyme and seleno-protein P. | [11,20,23,24] |
| | Anti-inflammatory effects | Se inhibits the activation of the NLRP3 inflammasome (leading to a reduction of IL-1β and IL-8. Reduces NF-κB pathway. | |
| **Zinc (Zn)** | Antioxidant Defense | Cofactor for superoxide dismutase (SOD). | [20,25,26] |
| | Synaptic Plasticity | Modulates NMDA receptor function. | |

IL: Interleukin; NF-kB: Nuclear Factor kappa-light-chain-enhancer of activated B cells; NLRP3: NOD-Like Receptor family, Pyrin domain containing 3; NMDA: N-methyl-D-aspartate receptor; VEGF: Vascular Endothelial Growth Factor.

been reported to be involved in a pathological cascade, leading to oxidative stress, synaptic dysfunction, and neural apoptosis after TBI [14]. To the best of our knowledge, there are no data on the concentration of TEs in living patients with TBI.

The aim of this exploratory study is first to assess the concentration of seven TEs: chromium (Cr), Cu, cobalt (Co), manganese (Mn), molybdenum (Mo), Se, and Zn in the extracellular fluid of the brain parenchyma (samples are provided by CMD technique) and systemic concentrations (serum) in a cohort of patients following severe TBI. Secondly, an investigation on any potential correlations between serum TEs levels and brain TEs concentrations is performed. Finally, we will explore any potential associations between TEs concentrations (brain/serum) and patient's neurological outcome at 12 months after trauma.

## Materials and methods

### Participants demographics and clinical details

Between March 2018 and August 2020, adult patients (≥18 years old) with severe TBI (defined by a Glasgow Coma Scale score < 9) admitted to the Department of Adult Intensive Care Medicine, Lausanne University Hospital (CHUV), Switzerland, were prospectively recruited.

The Glasgow Coma Scale (GCS) is a clinical tool used to assess the level of consciousness of patients with brain injury. It assesses three domains of responsiveness, including ocular, motor, and verbal components which are scored with 4, 6, and 5 categories, respectively. This score ranges between 3 (worst response) to 15 (best response). This score is a useful tool for assessing severity of injury, need of intensive monitoring and outcome [27].

We excluded patients with previous history of significant TBI, neurological handicap, and/or previous significant disability and/or psychiatric illness. Moribund patients, or for whom the clinical staff decides to suspend medical treatment are also excluded. This study follows on from the previously BIO-AX-TBI study (Developing and Validating Blood and Imaging Biomarkers of Axonal Injury Following Traumatic Brain Injury) [28].

For the current study, the inclusion was limited to adult patients who underwent cerebral multimodal monitoring with CMD in combination with serum sampling that were collected in the mornings within the first 3 days following ICU admission day. Samples were stored at −80°C.

All patients, next of kin or legally authorized representatives provided signed informed consent to the study approved by the local Ethical Committee (CER-VD 2017−01757).

## General patient management

Patients were treated according to international guidelines [29]. All patients underwent mechanical ventilation (aiming to keep $PaO_2$ and $PaCO_2$ at 90–100 mmHg and 35–40 mmHg, respectively) and sedation-analgesia (with infusion of propofol, at a maximal dose of 4 mg/kg/h, and sufentanil infusion, at a maximal dose of 20 µg/h). Cerebral perfusion pressure was maintained at 60–70 mmHg, with the use of vasopressors (norepinephrine) and isotonic fluids (aiming for euvolemia). Normoglycemia (target arterial blood glucose 6–8 mmol/L, with the use of continuous insulin infusion), normothermia (core body temperature < 37.5°C) and the administration within the first 24h a daily "*Stress profile*" (multi-TEs and multi-vitamin perfusion as described previously [30]) were part of standard care in our ICU.

All clinical data were recorded in a clinical information system (MetaVision®, IMDsoft)

## Cerebral microdialysis samples

CMD catheter (71 Microdialysis Catheter, M Dialysis® Stockholm, Sweden) was inserted in the operating room by trained neurosurgeons and placed into the frontal brain parenchyma (in visually normal subcortical white matter). A computed tomography (CT) scan of the brain was used to confirm the placement of CMD catheter. Its membrane was a 100kDa cut-off and was perfused with either artificial cerebrospinal fluid (CSF) or dextran via a pump (106 Microdialysis pump, M Dialysis® Stockholm, Sweden) at a constant rate of 0.3 µL/min and samples collected hourly.

Analysed CMD samples were concomitant to morning serum sampling, from 2 hours before serum collection to 7 hours after serum collection, pooled down, and quantified for specific TEs. Blood C-Reactive Protein (CRP) levels were measured by immuno-turbidimetry, as a systemic inflammation with CRP > 10 mg/L being associated with redistribution between tissues and low blood levels [31].

## Trace elements quantification

The concentrations of TEs Cr, Cu, Co, Mn, Mo, Se, and Zn were measured in CMD and serum samples by inductively coupled plasma system coupled to mass spectrometry (ICP-MS; 7700 Series; Agilent, Palo Alto) as previously described [32,33]. To our knowledge, there is no internal quality control for cerebral microdialysates. This is why the results of this study are based on validations carried out in the biological matrices classically available, i.e., plasma/serum, blood and urine. S1 Table contains the analytical parameters of the internal quality controls used to validate our methods.

The collected CMD fluid TEs amounts were corrected for TEs contamination of the respective perfusion fluid (i.e., artificial CSF, and dextran fluid which were analyzed separately).

## Statistical analysis

Data are expressed as median and interquartile range [25; 75] except when otherwise stated. For each variable, normality of data distribution was tested with the Shapiro-Wilk test. The data correlations were analysed with the non-parametric Spearman test. Data processing and statistical analyses were conducted using the Python programming language (Python Software Foundation, https://www.python.org/) and JMP 17 (JMP®, Cary, NC, USA) software, respectively. Statistical significance was set at $p < 0.05$.

## Results

### Samples and patient characteristics

The patient data are summarized in Table 2. A total of 281 hourly CMD samples were collected, corresponding to 48 morning serum collection time points in twenty severe TBI patients (6 female, 14 males) within three days following ICU admission. The

**Table 2. Patient clinical characteristics and outcome.**

| Case number | Age (years) M/F | Initial GCS | GOS-E at 12-month | LOS (days) | Marshall CT classification | Microdialysis catheter position |
|---|---|---|---|---|---|---|
| 1 | 47 M | 3 | 5 | 13 | Diffuse injury II | R Frontal |
| 2 | 51 M | 5 | 6 | 7 | Non evacuated mass lesion | L Frontal |
| 3 | 70 M | 6 | 1 | 3 | Non evacuated mass lesion | R Frontal |
| 4 | 22 F | 11 | 1 | 22 | Evacuated mass lesion | R Frontal |
| 5 | 53 M | 3 | LF | 6 | Diffuse injury II | R Frontal |
| 6 | 30 M | 7 | 6 | 18 | Diffuse injury II | L Frontal |
| 7 | 25 M | 4 | LF | 7 | Evacuated mass lesion | L Frontal |
| 8 | 19 M | 3 | 6 | 22 | Diffuse injury III | R Frontal |
| 9 | 31 M | 7 | 8 | 16 | Diffuse injury IV | L Frontal |
| 10 | 72 M | 5 | 1 | 13 | Diffuse injury IV | R Frontal |
| 11 | 70 F | 14 | LF | 12 | Diffuse injury II | L Frontal |
| 12 | 65 M | 3 | 1 | 15 | Non evacuated mass lesion | R Frontal |
| 13 | 35 F | 4 | LF | 9 | Diffuse injury II | R Frontal |
| 14 | 18 F | 6 | 1 | 16 | Diffuse injury IV | L Frontal |
| 15 | 53 M | 3 | 6 | 23 | Diffuse injury II | R Frontal |
| 16 | 47 M | 4 | 1 | 23 | Diffuse injury II | R Frontal |
| 17 | 34 M | 7 | 7 | 18 | Diffuse injury IV | R Frontal |
| 18 | 26 M | 3 | 8 | 28 | Diffuse injury II | R Frontal |
| 19 | 28 F | 5 | LF | 19 | Diffuse injury II | R Frontal |
| 20 | 22 F | 6 | 8 | 6 | Diffuse injury II | R Frontal |

M/F = Male/female, GCS = Glasgow Coma Scale, GOS-E: Glasgow Outcome Scale-Extended, LOS: Length of stay, CT: Computed Tomography, LF: Lost to follow-up, R = Right, L = Left.

median initial GCS was 5 [3;7] with a median age of 35 [25; 53] years old. Ten patients were classified as Diffuse Injury II, one as Diffuse Injury III, four as Diffuse Injury IV, two as Evacuated Mass Lesion and three as Non-Evacuated Mass Lesion according to the Marshall CT classification [34]. The CMD samples were collected using catheters placed in the frontal lobes (depending on whether the patient was right- or left-handed, or whether the frontal lobe was severely affected), 70% of the patients received CMD catheter in the right frontal lobe. Four patients (20%) experienced good recovery defined as a score of 7 or 8 at the Glasgow Outcome Scale Extended (GOSE) [35] and 5 patients (25%) were lost to follow up at 12 months (Table 2).

### Trace elements

Simultaneous samples of CMD and serum were not available for each timepoint for all patients: 12 were available on day 1, 17 on day 2, and 19 on day 3 (Fig 1). Available samples after the administration of *"Profil Stress"* were for 6 patients on day 1, 15 patients on day 2 and 18 patients on day 3. Potential contaminations: artificial CSF fluid contained only Se (0.25 µg/L), whereas the dextran perfusion fluid contained traces of all studied TEs: Cr (2.04 µg/L), Cu (1.01 µg/L), Co (0.052 µg/L), Mo (0.35 µg/L), Mn (0.99 µg/L), Se (0.45 µg/L) and Zn (5.79 µg/L). These data were used to correct the CMD samples for each TEs depending on the perfusion fluid. Results are presented in Table 3.

Fig 2 shows no statistically significant correlation between serum and CMD concentration for all the studied TEs except for Molybdenum (rho Spearman = 0.44, p-val = 0.005), after the administration of "*Stress Profile*". CRP levels were <10 mg/L in only 3 patients from first measure onwards. To note, CMD quantification of TE did not show any significant variations between the two hemispheres.

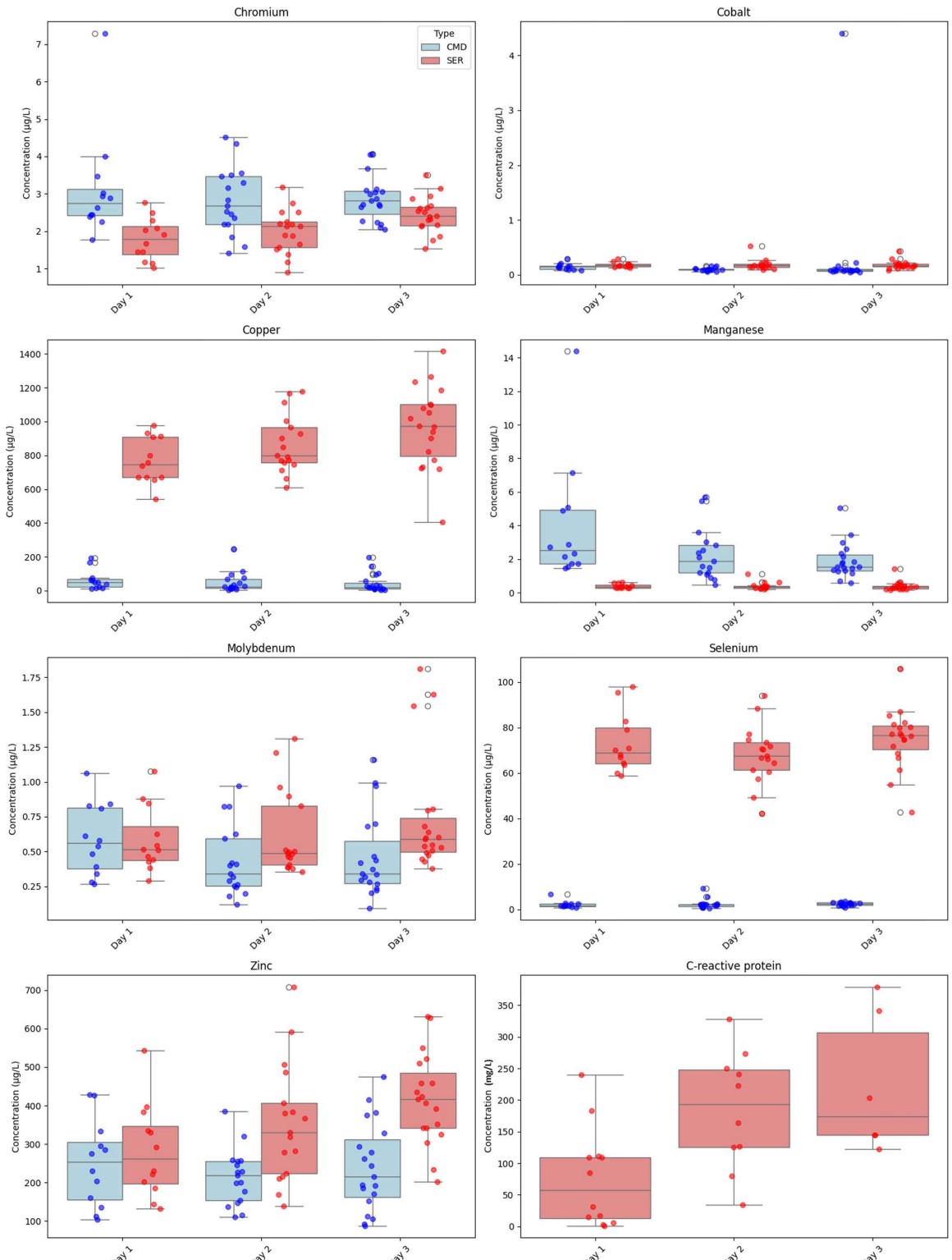

**Fig 1. Distribution of trace element levels in serum (SER) and in cerebral microdialysis fluid (CMD) according to time after intensive care unit admission (D: Day) for each trace element and C-Reactive Protein (CRP).**

**Table 3. Trace element quantification in cerebral microdialysis and serum samples after administration of "*Stress profile*".**

| Trace element, [µg/L] | Cerebral microdialysis | Serum |
|---|---|---|
| **Chromium** | | |
| Day 1 (n = 6) | 2.44 [2.24–2.92] | 2.18 [1.94–2.57] |
| Day 2 (n = 15) | 2.53 [2.19–3.30] | 2.14 [1.65–2.50] |
| Day 3 (n = 18) | 2.77 [2.26–3.06] | 2.45 [2.16–2.72] |
| **Cobalt** | | |
| Day 1 (n = 6) | 0.12 [0.09–0.16] | 0.17 [0.15–0.21] |
| Day 2 (n = 15) | 0.09 [0.08–0.10] | 0.17 [0.13–0.20] |
| Day 3 (n = 18) | 0.08 [0.06–0.10] | 0.16 [0.14–0.21] |
| **Copper** | | |
| Day 1 (n = 6) | 26.0 [12.4–51.6] | 705 [636–926] |
| Day 2 (n = 15) | 22.7 [15.8–76.6] | 792 [746–965] |
| Day 3 (n = 18) | 17.6 [11.0–38.4] | 971 [761–1100] |
| **Manganese** | | |
| Day 1 (n = 6) | 1.93 [1.65–2.43] | 0.32 [0.27–0.60] |
| Day 2 (n = 15) | 1.87 [1.17–2.81] | 0.33 [0.26–0.38] |
| Day 3 (n = 18) | 1.63 [1.34–2.41] | 0.32 [0.22–0.41] |
| **Molybdenum** | | |
| Day 1 (n = 6) | 0.37 [0.28–0.70] | 0.59 [0.42–0.90] |
| Day 2 (n = 15) | 0.40 [0.25–0.63] | 0.50 [0.39–0.90] |
| Day 3 (n = 18) | 0.36 [0.28–0.68] | 0.59 [0.50–0.80] |
| **Selenium** | | |
| Day 1 (n = 6) | 1.55 [1.30–2.39] | 67.8 [62.6–86.5] |
| Day 2 (n = 15) | 2.03 [0.85–2.22] | 68.8 [60.4–74.5] |
| Day 3 (n = 18) | 2.29 [1.81–3.08] | 76.5 [68.2–80.5] |
| **Zinc** | | |
| Day 1 (n = 6) | 148 [110–227] | 226 [141–355] |
| Day 2 (n = 15) | 201 [147–257] | 318 [215–383] |
| Day 3 (n = 18) | 205 [142–302] | 412 [338–471] |

Measures reported in median and 25–75 interquartile ranges.

## Discussion

To the best of our knowledge, this study is the first to present TEs data from the brain interstitial fluid collected by CMD in living patients using the analytical gold standard ICP-MS technique.

Essential TEs are elements present in very small amounts in the body with total amounts ranging from <1 mg to 5 g [36,37], capable of crossing the semi-permeable membrane of CMD. The choice of seven TEs (Cr, Co, Cu, Mn, Mo, Se, Zn) was based on their metabolic functions, and potential involvement in antioxidant defenses (Cu, Mn, Se, Zn), and glucose metabolism (Cr, Co, Mo, Zn) [38] at the cellular and mitochondrial level, two functions which are compromised in TBI [39].

In living patients, some TEs (Mn, Se, Zn, Cu, Fe) have been assessed in the CSF so far [40,41]. A major difference with previous brain status studies is that the TEs quantities were expressed as concentrations in microgram (µg) per dry brain mass (g), whereas our results are in µg per volume (mL) of fluid, making the comparison difficult [42].

No correlation was observed between serum and extracellular brain fluid for all TEs concentrations except one, Mo. The largest differences between serum and CMD levels were observed for Cu and Se, two TEs essential for antioxidant

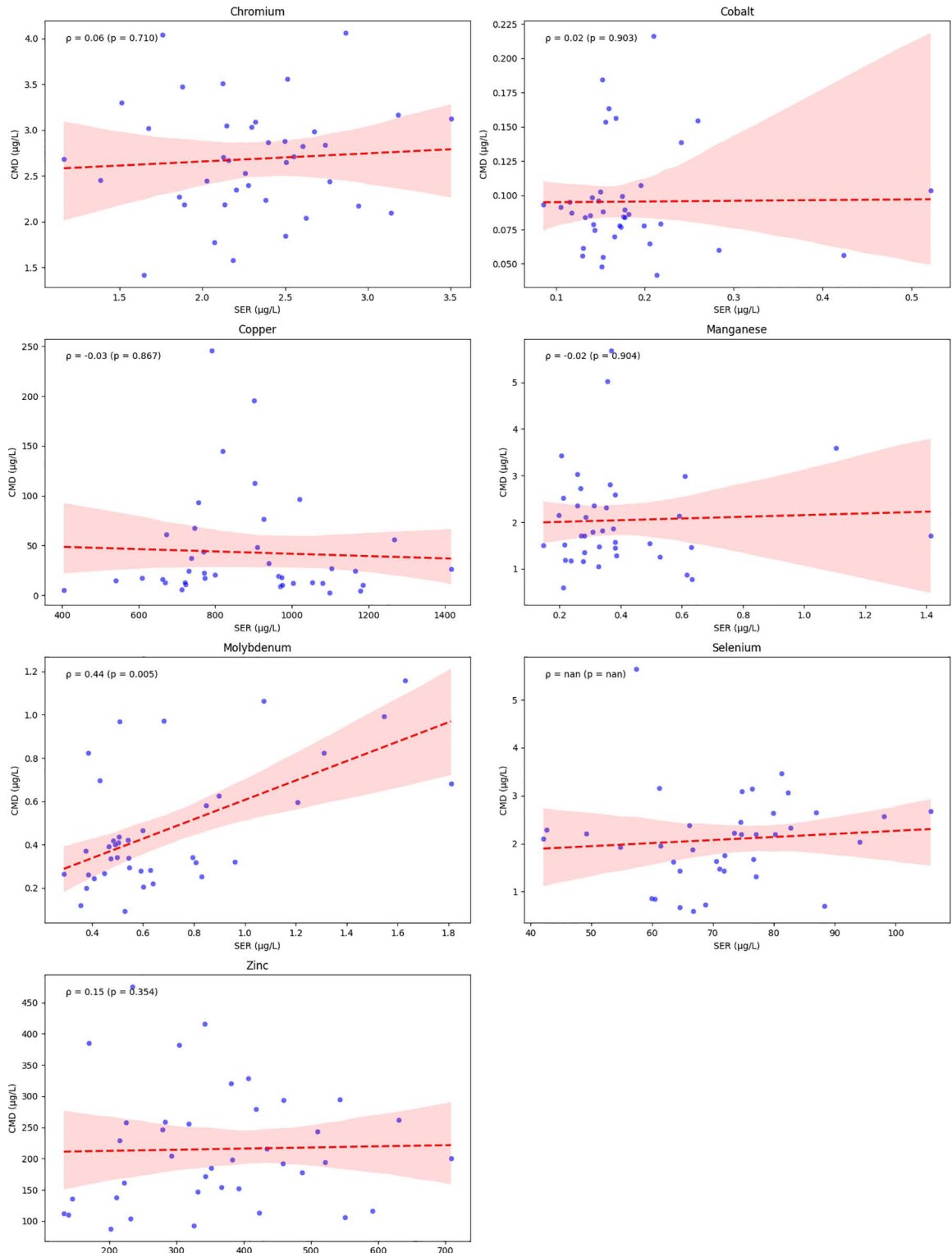

**Fig 2. Correlation between trace elements in serum and cerebral microdialysis fluid after administration of "*Stress profile*" (n = 39 observations).** Spearman correlation (ρ) between trace elements levels in serum (SER) and microdialysis fluid (CMD).

defense, followed by Cr and Zn, which have antioxidant defenses, and regulation of glucose levels. Serum levels were higher, likely reflecting the tight regulation and filtering role of the blood-brain barrier (BBB), as most trace elements (TEs) are not free in the serum but are bound to transport proteins—such as ceruloplasmin for copper or incorporated into proteins like selenoproteins for selenium—further suggesting a tightly controlled mechanism [43]. BBB passage is controlled by transcellular transport system subject to regulation and saturation. This is a new finding that deserves further research.

The CMD catheter position may be a confounder, as regional and even inter-hemispheric variations of concentrations have been described. Indeed, significant interhemispheric differences in Cu concentration have been reported [44,45] as well as differences in Se, Zn concentrations between different regions of normal human brains (hippocampus, cerebellum, frontal, parietal, temporal, and occipital) [45].

The TEs status and resultant serum level depend on a wide range of factors, such as age, sex, diet, Body Mass Index, sociodemographic and disease status and are therefore variable [46–49]. In our cohort, all our patients received the same nutritional support in the ICU. However, nutritional intakes were different before hospital admission. A study investigating TEs levels in brain autopsy tissues from patients of various ages ranging from premature infants to 85 years old showed some age-dependent changes. Co concentration in the brain increases until the age of 79 years old and then declined, while Mn and Cr were similar across ages. Se and Zn levels seems to be constant in adults [50], which may be related to their essential antioxidant role. The current study included patients aged between 22–70 years old, reducing previously mentioned age impact.

The strength of this study is that it is the first to measure TEs in living patients and the simultaneous collection of serum and CMD samples.

Among the limitations of the study, the small sample size explained by the limited volumes collected and those remaining after the primary study's BIO-AX-TBI analytical outwork [28]. The second limitation is related to the first, with the level of inflammation (CRP determination) not being systematically assessed, with missing CRP values. Indeed, micronutrient serum levels decrease as soon as CRP exceeds 10 mg/L due to the cytokine mediated redistribution. The presence of inflammation in the context of trauma or acquired infection can alter the levels of TEs in serum. Duncan et al. [31] showed that Se and Zn decrease proportionally to the inflammation reflected by increasing CRP, while Cu increases. The low serum Se and Zn levels are explained by this fact. An important limitation is the unavailability of Fe due to the low volume of fluid available and would have been important in the assessment of oxidative metabolism, Fe being prooxidant.

## Conclusion

In conclusion, the concentrations of seven TEs (Cr, Cu, Co, Mn, Mo, Se, and Zn) collected simultaneously in serum and extracellular brain fluid during the first week after severe TBI were successfully quantified. However, serum and brain interstitial fluid samples were not correlated. The results of this original approach in living humans can serve as a basis for further research exploring TEs dynamics in the brain and thus a better understanding of TEs variations in patients with an ABI.

## Supporting information

**S1 Table. Analytical parameters of the internal quality controls (ClinCheck Controls, Recipe) used to validate our methods in two matrices (serum and urine).** Abbreviations: LOD: detection limit; LOQ: quantification limit; CVr: repeatability; CVR: reproductibility.
(DOCX)

**S2 Table. Dataset.** Abbreviations: M: male; F:female; GCS: Glasgow Coma Scale; GOSE: Glasgow Coma Scale Extended; MD: microdialysis fluid; BD: blood serum. (Quantification in µg/L for all samples).
(XLSX)

## Author contributions

**Conceptualization:** Mette M. Berger, Nawfel Ben-Hamouda.

**Data curation:** Samia Abed-Maillard.

**Formal analysis:** Adriano Bernini, Sébastien Lenglet, Roy Thomas Daniel, Mahmoud Messerer, Jean-Daniel Chiche, Marc Augsburger.

**Funding acquisition:** Samia Abed-Maillard.

**Investigation:** Sébastien Lenglet, Roy Thomas Daniel, Mahmoud Messerer, Mauro Oddo, Marc Augsburger.

**Methodology:** Adriano Bernini, Sébastien Lenglet, Mette M. Berger, Marc Augsburger, Nawfel Ben-Hamouda.

**Project administration:** Samia Abed-Maillard, Nawfel Ben-Hamouda.

**Software:** Adriano Bernini.

**Supervision:** Jean-Daniel Chiche, Nawfel Ben-Hamouda.

**Validation:** Mette M. Berger, Nawfel Ben-Hamouda.

**Visualization:** Mauro Oddo.

**Writing – original draft:** Adriano Bernini, Mette M. Berger, Nawfel Ben-Hamouda.

**Writing – review & editing:** Adriano Bernini, Sébastien Lenglet, Mette M. Berger, Samia Abed-Maillard, Roy Thomas Daniel, Mahmoud Messerer, Mauro Oddo, Jean-Daniel Chiche, Marc Augsburger, Nawfel Ben-Hamouda.

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
