## [Decision Letter · Decision Letter 0]

Feb 09 2025

Dear Dr. BEN-HAMOUDA,

Thank you for submitting your manuscript to PLOS ONE. After careful consideration, we feel that it has merit but does not fully meet PLOS ONE’s publication criteria as it currently stands. Therefore, we invite you to submit a revised version of the manuscript that addresses the points raised during the review process.

We look forward to receiving your revised manuscript.

Kind regards,

Kenji Tanigaki, Ph.D., M.D.

Academic Editor

PLOS ONE

Journal Requirements:

2. In the online submission form, you indicated that “The data underlying this study are available from the corresponding author on reasonable request.”

Reviewers' comments:

Reviewer's Responses to Questions

**Comments to the Author**

1. Is the manuscript technically sound, and do the data support the conclusions?

Reviewer #1: Yes

Reviewer #2: Partly

2. Has the statistical analysis been performed appropriately and rigorously?

Reviewer #1: Yes

Reviewer #2: Yes

3. Have the authors made all data underlying the findings in their manuscript fully available?

Reviewer #1: Yes

Reviewer #2: Yes

4. Is the manuscript presented in an intelligible fashion and written in standard English?

Reviewer #1: Yes

Reviewer #2: Yes

Reviewer #1: Brain microdialysis to assess trace elements dynamics in traumatic brain injury: an exploratory study

The manuscript should be revised for minor linguistic, grammatical and style errors.

The manuscript should be revised for proper use of abbreviations

The neuroprotective pathways should be discussed deeply in introduction and discussion sections. The following reference might be helpful:

https://doi.org/10.1016/j.biopha.2021.111729

https://doi.org/10.3390/nu12041028

https://doi.org/10.3389/fphar.2020.619024

Reviewer #2: Manuscript ID: PONE-D-24-37719

Title: Brain microdialysis to assess trace elements dynamics in traumatic brain injury: an exploratory study

The article submitted by Nawfel Ben-Hamouda et al. to PLOS ONE entitled "Brain microdialysis to assess trace elements dynamics in traumatic brain injury: an exploratory study" explores trace element (TE) concentrations in the brain and serum of severe traumatic brain injury (TBI) patients using cerebral microdialysis and ICP-MS. Significant differences were observed, with copper (Cu) and selenium (Se) levels 8–10 times higher in serum than in brain interstitial fluid, indicating tight blood-brain barrier regulation. No correlation was found between five other TE levels in serum and brain fluid, highlighting independent systemic and brain-specific dynamics. Elements like Cu, Se and zinc (Zn), crucial for antioxidant defense, showed distinct distributions, reflecting localized metabolic demands. This exploratory study provides foundational data for understanding TE roles in acute brain injury and potential therapeutic applications.

The work described in this article is novel, relevant to the readership of PLOS ONE, and well-structured overall. However, several sections of the manuscript need further clarification before publication. My current suggestion is to consider the manuscript for publication after Major Revision as outlined in the following details:

General comments:

1. The variability in catheter placement introduces significant confounding factors, as TE concentrations vary across brain regions. For example, inter-hemispheric differences in copper and regional disparities in Se and Zn are well-documented, but this study does not control for or stratify based on catheter location. Without a standardized placement protocol or regional analysis, observed differences may reflect sampling biases rather than true systemic dynamics. This limitation complicates the interpretation of results and reduces the generalizability of findings, as TE dynamics may differ significantly depending on the sampled brain region. Although sample size is limited, some exploration into this direction might be worth trying.

2. The incomplete measurement of C-reactive protein (CRP) limits the ability to account for systemic inflammation, a critical factor affecting TE dynamics. Inflammatory responses following TBI can cause TE redistribution, leading to decreased serum levels of Se and Zn and increased Cu levels. Without consistent CRP data, it is difficult to determine whether observed TE changes are due to TBI pathology or inflammation. Systematic measurement of CRP and other inflammatory markers would provide essential context for interpreting TE levels in both serum and brain interstitial fluid.

3. Throughout the manuscript: there appears to be a mix of units used for TE concentrations. E.g., in table 2, the authors state that the seven TEs measured in serum and interstitial brain fluid are in “µg/ml”, while in the text they mostly refer to “ng/ml”, but providing comparable values. The concentration units in table 2 are probably by a factor of ~1,000 × too high. Please carefully check and revise all units. For clarity, I suggest using the same units throughout the manuscript.

4. Throughout the manuscript, there appear several abbreviations that are never used afterwards (e.g., ABI in line 70), or not explained at all (e.g., CSF in line 214). Please doublecheck all abbreviations.

5. Throughout the manuscript (text, figures, table), the authors refer to serum and blood samples and abbreviate them as BD and SER. These appear to be mixed up and it is difficult for the reader to follow. As I understand, the authors analyzed TEs in serum, but not blood. Please clarify.

6. It would be helpful for the readership to briefly introduce the Glasgow Coma Scale score, the range, and what the score reflects.

7. Paragraph “General patient management” (page 5): The authors should consider carefully providing any TEs that might be administered as part of the described procedures (e.g., Zn in multi-vitamin perfusion or isotonic fluids). These perfusions likely alter the patients’ initial TE levels and might explain some of the null correlations observed.

8. Paragraph “Trace elements quantification” (page 6): The authors need to present the method used for the analysis of the 7 TEs in serum and CMD. The current reference [20] does not describe the analytical procedures for the analysis of any of these two biospecimen. There also needs to be a section added on the performed quality control and quality assurance (QC/QA) including the presentation of basic QC/QA parameters such as accuracy, precision, detection limits, etc. Without this essential information, this manuscript and concomitant conclusions cannot be adequately evaluated.

9. Paragraph “Data availability” (page 7): The authors should state the source of presented patient data (e.g., age, GCS, CT-classifications).

10. Paragraph “Trace elements” (pages 8-9): There appear to be several deviations between the provided text, figure 1 and table 2. The authors state that: “…serum and CMD levels do not different much for Co, Mo and Mn, and moderate difference are observable for Cr and Zn,…”. Comparing the median concentrations in table 2 and figure 1, it appears that Co and Mn show a factor of 2-5 × difference between serum and interstitial fluid, while Cr and Zn only feature none at all to up to 3 × differences. Also, the provided factor of 8-10 × between serum and interstitial fluid for Cu and Se appear to be much larger (16-57 ×). The authors also state that: “Fig 2 shows no correlation between serum and CMD concentration for the seven studied TEs after the administration of “Stress Profile”.” Depending on the authors’ definitions of no, weak, moderate, and strong correlations, in my experience, a Spearman correlation coefficient of |p|>0.5 (as observed for Cr, Se, and Zn) should be considered at least as weak or moderate. In addition, Figure 2 misses the plot for Mo. Please carefully doublecheck and revise this section, figure 1 and table 2.

11. Table 2: Several of the provided medians are not within the provided IQR, or IQR is presented from high to low (e.g., Co / Day 2). Consider reducing the number of significant figures for the presented concentrations to three or four (e.g., Zn / Day 1: from 247.276 to 274). Several TE concentrations in this table appear very high compared to normal serum and interstitial fluids compared to e.g., NHANES and other published data, even considering that the correct units should likely be ng/mL rather that the presented µg/mL (e.g., Cr), while one might expect others (e.g., Zn) to be very different between serum and interstitial brain fluid (Kapaki et al. 1989 Zinc, copper and magnesium concentration in serum and CSF of patients with neurological disorders. Acta Neurol Scand.79(5):373-8. doi: 10.1111/j.1600-0404.1989.tb03803.x. PMID: 2545071). It is possible, that the analytical procedure has a significant influence on these data, which highlights the importance of providing a detailed method and QC/QA data. For Mo, please replace the “0” values on Days 2 and 3 with the method’s detection limit (e.g., “<0.05”).

12. Lines 252-253: The authors state that Fe was a desirable TE in their study, but that it was not possible to analyze it due to the low sample volumes available. Please describe how much sample was available for each biospecimen. Particularly for serum, this sounds implausible (serum samples usually yield several mL per blood draw). However, even if sample volume was limited, the huge advantage of ICP-MS is that this method can quasi-simultaneously quantify dozens of TEs, and therefore, it should have been possible to measure Fe together with the seven other TEs using the same analytical procedure. The additional sample amount needed is negligible. This omission represents a missed opportunity to comprehensively evaluate oxidative metabolism.

Specific comments:

1. Abstract: It would be good to mention the number of patients examined.

2. Line 60: Please add “…recovery…” from what.

3. Line 86: Delete “were”?

4. Lines 88-89: In the last sentence of this paragraph, the authors state that: “To our knowledge, there are no data about TEs concentrations in patients with TBI.” Are the authors referring to TE data particularly in brain tissues and cerebrospinal fluid, or to any biospecimen of TBI patients?

5. Line 96: do the authors mean “…neurological outcome at 12 months past TBI”?

6. Line 101: Please add the total number of patients examined.

7. Line 107: Please provide full name of the BIO-AX-TBI study.

8. Lines 131-135: Where serum collection containers tested for TE background? Please provide the method for the measurement of CRP.

9. Table 2: Add number of patients for each TE and day per median (IQR).

10. Figure 2: Please add the number of presented observations.

11. Line 208: Please replace “metals” with something like “elements” (e.g., Se is not a metal).

12. Line 214: Change “SI” to “Si”; and define “CSF”.

13. Line 217: Change “µL” to “mL”.

14. Line 218: Following up on my earlier comment on the definition of no, weak, moderate, and strong correlations, this sentence should be revised.

15. Line 220: The presented factors of 8 to 10 fold higher should be doublechecked. They should be much higher according to the presented median values in table 2.

16. Line 222: Some TEs, such as Se are not considered to be “transported with proteins”, but rather are part of the proteins’ polypeptide chains (e.g., as selenoprotein P contains the amino acid selenocysteine). Please consider rephrasing.

17. Lines 224-225: Please doublecheck the factors of TE differences between sample matrices. Co and Mn feature larger differences than Cr and Zn according to table 2 and figure 1 (consider enlarging the y-axis scale for Co to better see the differences).

18. Line 233: The authors state: “The TEs status and resultant serum level depend on diet and age and are therefore variable.” There are many other factors also correlated with TE status (e.g., sex, BMI, socio-demographics, disease status). Please consider adding or mention there are more confounders on TE status in serum.

**Do you want your identity to be public for this peer review?** For information about this choice, including consent withdrawal, please see our Privacy Policy

Reviewer #1: No

Reviewer #2: **Yes: ** Ronald A. Glabonjat

---

## [Author Response · Author response to Decision Letter 1]

29 Apr 2025

Point-by-point responses to the comments

Reviewer #1: Brain microdialysis to assess trace elements dynamics in traumatic brain injury: an exploratory study

The manuscript should be revised for minor linguistic, grammatical and style errors.

The manuscript should be revised for proper use of abbreviations

The neuroprotective pathways should be discussed deeply in introduction and discussion sections. The following reference might be helpful:

https://doi.org/10.1016/j.biopha.2021.111729

https://doi.org/10.3390/nu12041028

https://doi.org/10.3389/fphar.2020.619024

Response: Many thanks for the revision. We have added Table 1 to detail the neuroprotective pathways of trace elements in the traumatic brain injury and we have corrected errors and the use of abbreviations.

Reviewer #2: Manuscript ID: PONE-D-24-37719

Title: Brain microdialysis to assess trace elements dynamics in traumatic brain injury: an exploratory study

The article submitted by Nawfel Ben-Hamouda et al. to PLOS ONE entitled "Brain microdialysis to assess trace elements dynamics in traumatic brain injury: an exploratory study" explores trace element (TE) concentrations in the brain and serum of severe traumatic brain injury (TBI) patients using cerebral microdialysis and ICP-MS. Significant differences were observed, with copper (Cu) and selenium (Se) levels 8–10 times higher in serum than in brain interstitial fluid, indicating tight blood-brain barrier regulation. No correlation was found between five other TE levels in serum and brain fluid, highlighting independent systemic and brain-specific dynamics. Elements like Cu, Se and zinc (Zn), crucial for antioxidant defense, showed distinct distributions, reflecting localized metabolic demands. This exploratory study provides foundational data for understanding TE roles in acute brain injury and potential therapeutic applications.

The work described in this article is novel, relevant to the readership of PLOS ONE, and well-structured overall.

Response: Many thanks for these positive remarks and for your revision

However, several sections of the manuscript need further clarification before publication. My current suggestion is to consider the manuscript for publication after Major Revision as outlined in the following details:

General comments:

1. The variability in catheter placement introduces significant confounding factors, as TE concentrations vary across brain regions. For example, inter-hemispheric differences in copper and regional disparities in Se and Zn are well-documented, but this study does not control for or stratify based on catheter location. Without a standardized placement protocol or regional analysis, observed differences may reflect sampling biases rather than true systemic dynamics. This limitation complicates the interpretation of results and reduces the generalizability of findings, as TE dynamics may differ significantly depending on the sampled brain region. Although sample size is limited, some exploration into this direction might be worth trying.

Response: As presented in the results part, patient management protocol states to insert the CMD catheter in the frontal brain parenchyma, in visually normal subcortical white matter. In our cohort, we had 6 patients with the CMD catheter position on the left frontal brain parenchyma and 14 on the right frontal brain parenchyma. No significant differences have been observed between the 2 hemispheres.

2. The incomplete measurement of C-reactive protein (CRP) limits the ability to account for systemic inflammation, a critical factor affecting TE dynamics. Inflammatory responses following TBI can cause TE redistribution, leading to decreased serum levels of Se and Zn and increased Cu levels. Without consistent CRP data, it is difficult to determine whether observed TE changes are due to TBI pathology or inflammation. Systematic measurement of CRP and other inflammatory markers would provide essential context for interpreting TE levels in both serum and brain interstitial fluid.

Response: Totally agree, this was reported in the limitations of the study.

3. Throughout the manuscript: there appears to be a mix of units used for TE concentrations. E.g., in table 2, the authors state that the seven TEs measured in serum and interstitial brain fluid are in “µg/ml”, while in the text they mostly refer to “ng/ml”, but providing comparable values. The concentration units in table 2 are probably by a factor of ~1,000 × too high. Please carefully check and revise all units. For clarity, I suggest using the same units throughout the manuscript.

Response: Thank you, we corrected that and used the same unit throughout the manuscript. The quantifications were reported in µg/L. The values reported in table 3 were corrected and reported now in µg/L.

4. Throughout the manuscript, there appear several abbreviations that are never used afterwards (e.g., ABI in line 70), or not explained at all (e.g., CSF in line 214). Please doublecheck all abbreviations.

Response: Thank you. We corrected them.

5. Throughout the manuscript (text, figures, table), the authors refer to serum and blood samples and abbreviate them as BD and SER. These appear to be mixed up and it is difficult for the reader to follow. As I understand, the authors analyzed TEs in serum, but not blood. Please clarify.

Response: Yes, indeed we quantify TEs in serum and this has been corrected throughout the manuscript, table and figures.

6. It would be helpful for the readership to briefly introduce the Glasgow Coma Scale score, the range, and what the score reflects.

Response: We added that in the “Participants demographics and clinical details” section.

7. Paragraph “General patient management” (page 5): The authors should consider carefully providing any TEs that might be administered as part of the described procedures (e.g., Zn in multi-vitamin perfusion or isotonic fluids). These perfusions likely alter the patients’ initial TE levels and might explain some of the null correlations observed.

Response: All the patients received the same multi-TEs and multivitamin perfusion within 24h of admission to the ICU. No samples of TEs were analyzed before this perfusion.

8. Paragraph “Trace elements quantification” (page 6): The authors need to present the method used for the analysis of the 7 TEs in serum and CMD. The current reference [20] does not describe the analytical procedures for the analysis of any of these two biospecimen. There also needs to be a section added on the performed quality control and quality assurance (QC/QA) including the presentation of basic QC/QA parameters such as accuracy, precision, detection limits, etc. Without this essential information, this manuscript and concomitant conclusions cannot be adequately evaluated.

Response: Thank you for your comments, which require further development. Indeed, reference 20 (Perrais et al., 2023) contains many analytical parameters which are similar to those used for the measurement of trace element concentrations in this study. To complete the analytical procedure for biological samples, we have added a second reference (Perrais et al., 2024) to a recent work on the measurement of trace element concentrations (in plasma and urine) in a population cohort of around 1100 participants:

Reference values for plasma and urine trace elements in a Swiss population-based cohort. Perrais M et al. Clin Chem Lab Med, 2024, 62(11):2242-2255.

S1 Table (we added supporting information file) of this study contains the analytical parameters of the internal quality controls (ClinCheck Controls, Recipe) used to validate our methods in these two matrices (serum and urine).

Based on this new reference (Perrais et al., 2024), you will find below the parameters relative to the trace elements analyzed in serum in the present study submitted to you:

Trace Element LOD (µg/L) LOQ (µg/L) CVr(%) CVR(%)

Chromium Cr 0.189 0.559 3.30% 4.50%

Manganese Mn 0.123 0.377 3.90% 7.90%

Cobalt Co 0.003 0.015 1.80% 4.50%

Copper Cu 1.147 3.691 1.70% 3.10%

Zinc Zn 0.421 5.536 1.20% 3.90%

Selenium Se 0.614 1.680 1.40% 2.50%

Molybdenum Mo 0.023 0.074 2.00% 2.80%

LOD : detection limit

LOQ : quantification limit

CVr : repetability

CVR : reproductibility

To our knowledge, there is no internal quality control for cerebral microdialysates. This is why the results of this study are based on validations carried out in the biological matrices classically available, i.e. plasma/serum, blood and urine.

In addition to the internal quality controls used (ClinChek Control, Recipe) during acquisition, we also participate in two external quality control programs for different matrices (blood, serum, urine, hair):

- QMEQAS (Quebec Multielement External Quality Assessment), frequency: 3x per year

https://www.inspq.qc.ca/en/ctq/eqas/qmeqas/description

- LAMP (Lead and Multielement Proficiency Program), frequency: 4x a year

https://www.cdc.gov/laboratory-quality-assurance/php/inorganic-elements/lamp.html

All trace elements analyzed in this study (Cr, Co, Mn, Mo, Cu, Zn, Se) are included in the external quality control programs mentioned.

Finally, our laboratory's trace element analysis by ICP-MS is accredited by the Swiss Accreditation Service (STS 0448) to ISO 17025.

9. Paragraph “Data availability” (page 7): The authors should state the source of presented patient data (e.g., age, GCS, CT-classifications).

Response: All clinical data were recorded in a clinical information system. We added it.

10. Paragraph “Trace elements” (pages 8-9): There appear to be several deviations between the provided text, figure 1 and table 2. The authors state that: “…serum and CMD levels do not different much for Co, Mo and Mn, and moderate difference are observable for Cr and Zn,…”. Comparing the median concentrations in table 2 and figure 1, it appears that Co and Mn show a factor of 2-5 × difference between serum and interstitial fluid, while Cr and Zn only feature none at all to up to 3 × differences. Also, the provided factor of 8-10 × between serum and interstitial fluid for Cu and Se appear to be much larger (16-57 ×). The authors also state that: “Fig 2 shows no correlation between serum and CMD concentration for the seven studied TEs after the administration of “Stress Profile”.” Depending on the authors’ definitions of no, weak, moderate, and strong correlations, in my experience, a Spearman correlation coefficient of |p|>0.5 (as observed for Cr, Se, and Zn) should be considered at least as weak or moderate. In addition, Figure 2 misses the plot for Mo. Please carefully doublecheck and revise this section, figure 1 and table 2.

Response: we removed those affirmations are they were out of the scope of the study and is subject to interpretability.

11. Table 2: Several of the provided medians are not within the provided IQR, or IQR is presented from high to low (e.g., Co / Day 2). Consider reducing the number of significant figures for the presented concentrations to three or four (e.g., Zn / Day 1: from 247.276 to 274). Several TE concentrations in this table appear very high compared to normal serum and interstitial fluids compared to e.g., NHANES and other published data, even considering that the correct units should likely be ng/mL rather that the presented µg/mL (e.g., Cr), while one might expect others (e.g., Zn) to be very different between serum and interstitial brain fluid (Kapaki et al. 1989 Zinc, copper and magnesium concentration in serum and CSF of patients with neurological disorders. Acta Neurol Scand.79(5):373-8. doi: 10.1111/j.1600-0404.1989.tb03803.x. PMID: 2545071). It is possible, that the analytical procedure has a significant influence on these data, which highlights the importance of providing a detailed method and QC/QA data. For Mo, please replace the “0” values on Days 2 and 3 with the method’s detection limit (e.g., “<0.05”).

Response: We verified all values and corrected the table (table 3) for the contamination reported with particular attention to the units.

12. Lines 252-253: The authors state that Fe was a desirable TE in their study, but that it was not possible to analyze it due to the low sample volumes available. Please describe how much sample was available for each biospecimen. Particularly for serum, this sounds implausible (serum samples usually yield several mL per blood draw). However, even if sample volume was limited, the huge advantage of ICP-MS is that this method can quasi-simultaneously quantify dozens of TEs, and therefore, it should have been possible to measure Fe together with the seven other TEs using the same analytical procedure. The additional sample amount needed is negligible. This omission represents a missed opportunity to comprehensively evaluate oxidative metabolism.

Response:

Brain interstitial fluid volumes are of the order of a few microliters. Our quantification methods are validated for the following elements: Cr, Co, Mn, Mo, Cu, Zn, Se. In order to quantify iron, we would have had to carry out a second independent calibration. Due to the very low volume of material available, it was impossible to prepare a second series of samples.

As you point out, ICP-MS technology enables multi-element analysis, particularly of serum, due to the larger sample volumes involved. The measurement of iron concentrations could have provided additional information on oxidative stress. However, as this information was not available for cerebral microdialysis samples, we found it difficult to interpret iron concentrations based on results from a single matrix. In addition, since ICP-MS technology is a destructive analysis, the iron concentration measured would be the total, i.e. the sum of the free and bound forms. We will take this point into account for a future study and attempt to measure the different forms of iron in serum using other methodologies.

Specific comments:

1. Abstract: It would be good to mention the number of patients examined.

Done

2. Line 60: Please add “…recovery…” from what.

Corrected.

3. Line 86: Delete “were”?

Corrected.

4. Lines 88-89: In the last sentence of this paragraph, the authors state that: “To our knowledge, there are no data about TEs concentrations in patients with TBI.” Are the authors referring to TE data particularly in brain tissues and cerebrospinal fluid, or to any biospecimen of TBI patients?

Modified. Here we mean “in living patients with TBI”.

5. Line 96: do the authors mean “…neurological outcome at 12 months past TBI”?

Modified.

6. Line 101: Please add the total number of patients examined.

Done.

7. Line 107: Please provide full name of the BIO-AX-TBI study.

Added.

8. Lines 131-135: Where serum collection containers tested for TE background? Please provide the method for the measurement of CRP.

TE are tested at the Unit of Forensic Toxicology and Chemistry (University Center of Legal Medicine, Lausanne-Geneva, Switzerland)

Immuno-turbidimetry.

9. Table 2: Add number of patients for each TE and day per median (IQR).

Done

10. Figure 2: Please add the number of presented observations.

Added.

11. Line 208: Please replace “metals” with something like “elements” (e.g., Se is not a metal).

Modified in “elements”

12. Line 214: Change “SI” to “Si”; and define “CSF”.

Modified. CSF has been defined in the manuscript (Cerebral microdialysis samples section).

13. Line 217: Change “µL” to “mL”.

Done

14. Line 218: Following up on my earlier comment on the definition of no, weak, moderate, and strong correlations, this sentence should be revised.

We corrected into statistically significant

15. Line 220: The presented factors of 8 to 10 fold higher should be doublechecked. They should be much higher according to the presented median values in table 2.

Values were corrected and we removed this fold factor as it was subject to interpretability.

16. Line 222: Some TEs, such as Se are not considered to be “transported with proteins”, but rather are part of the proteins’ polypeptide chains (e.g., as selenoprotein P contains the amino acid selenocysteine). Please consider rephrasing.

Modified.

---

## [Decision Letter · Decision Letter 1]

Brain microdialysis to assess trace elements dynamics in traumatic brain injury: an exploratory study

PONE-D-24-37719R1

Dear Dr. BEN-HAMOUDA,

We’re pleased to inform you that your manuscript has been judged scientifically suitable for publication and will be formally accepted for publication once it meets all outstanding technical requirements.

Kind regards,

Kenji Tanigaki, Ph.D., M.D.

Academic Editor

PLOS ONE

Additional Editor Comments (optional):

Reviewers' comments:

Reviewer's Responses to Questions

**Comments to the Author**

Reviewer #1: All comments have been addressed

Reviewer #2: All comments have been addressed

2. Is the manuscript technically sound, and do the data support the conclusions?

Reviewer #1: Yes

Reviewer #2: Yes

3. Has the statistical analysis been performed appropriately and rigorously?

Reviewer #1: Yes

Reviewer #2: Yes

4. Have the authors made all data underlying the findings in their manuscript fully available?

Reviewer #1: Yes

Reviewer #2: Yes

5. Is the manuscript presented in an intelligible fashion and written in standard English?

Reviewer #1: Yes

Reviewer #2: Yes

Reviewer #1: Manuscript Number PONE-D-24-37719R1

Brain microdialysis to assess trace elements dynamics in traumatic brain injury: an exploratory study

As the authors addressed the reviewer comments, I suggest acceptance of the manuscript.

Reviewer #2: Thank you for your careful consideration of all raised points. The only last gentle suggestion I have is regarding the number of significant figures presented (e.g., table 3) and to reduce to no more than maybe 3 (e.g., for Zn). This is a beautiful and important piece of work with high significance to the field.

**Do you want your identity to be public for this peer review?** For information about this choice, including consent withdrawal, please see our Privacy Policy

Reviewer #1: No

Reviewer #2: **Yes: ** Ronald A. Glabonjat

---

## [Editor Report · Acceptance letter]

PONE-D-24-37719R1

PLOS ONE

Dear Dr. BEN-HAMOUDA,

I'm pleased to inform you that your manuscript has been deemed suitable for publication in PLOS ONE. Congratulations! Your manuscript is now being handed over to our production team.

Kind regards,

on behalf of

Dr. Kenji Tanigaki

Academic Editor

PLOS ONE